# Assessing the clinical relevance of point-of-care ultrasound for hospitalists: Influence on clinical reasoning and decision-making

Maira Dias Souza[1]*, Hassan Rahhal[1,2], Iolanda F. L. C. Tibério[3]

1 Departamento de Clínica Médica, Instituto Central, Hospital das Clínicas da Faculdade de Medicina da Universidade de São Paulo, Faculdade de Medicina, Universidade de São Paulo, São Paulo, São Paulo, Brazil, 2 Faculdade de Medicina, Universidade de Sao Caetano do Sul, São Paulo, São Paulo, Brazil, 3 Faculdade de Medicina da Universidade de São Paulo, Universidade de São Paulo, São Paulo, São Paulo, Brazil

* mairadiassouza@gmail.com

## Abstract

### Introduction

Point-of-care ultrasonography (POCUS) is increasingly recognized in internal medicine, yet its influence on hospitalists' clinical reasoning remains underexplored.

### Objectives

This study aimed to evaluate the influence of POCUS on diagnostic hypotheses, management strategies, and confidence levels among hospitalists in a transitional care unit with limited access to advanced imaging.

### Methods

Prospective descriptive study in a transitional care unit enrolling 19 hospitalists. During routine care, clinicians used POCUS at their discretion and, immediately afterward, completed a structured form capturing the clinical question, pre- and post-POCUS diagnostic hypotheses, confidence levels, and management decisions. Per-encounter changes in these parameters were summarized descriptively.

### Results

POCUS led to a change in the primary diagnostic hypothesis in 37% (38/104) of clinical encounters and altered management in 39% (41/104) of cases. Diagnostic confidence increased in 46% (48/104) of instances. Even when the primary hypothesis remained unchanged (62% − 66/104), POCUS strengthened diagnostic confidence in 36% (24/66) of these cases and aided in ruling out alternative diagnoses in 62% (41/66). Notably, POCUS had no discernible influence in only 13% (14/104) of cases.

**Data availability statement:** All relevant data are within the paper and its Supporting Information files.

**Funding:** We would also like to acknowledge that this study was financially supported by the Fundação de Amparo à Pesquisa do Estado de São Paulo (FAPESP) through the Thematic Project (Grant nº 2018/02537-05). The funders had no role in study design, data collection and analysis, decision to publish, or preparation of the manuscript.

**Competing interests:** The authors have declared that no competing interests exist.

## Conclusion

This study suggests that POCUS may influence clinical reasoning and decision-making among hospitalists in a setting with limited imaging resources. POCUS frequently led to changes in diagnostic hypotheses and management plans, and often increased diagnostic confidence, even when the initial hypothesis was maintained. These findings suggest that POCUS may play a supportive role in bedside assessment and patient management in internal medicine, underscoring the need for further research and structured training before broader integration.

## Introduction

Point-of-care ultrasonography (POCUS) has been widely adopted across various medical specialties as a rapid, bedside imaging tool that enhances diagnostic accuracy and clinical decision-making [1–5]. Its use is well established in critical care settings, where it has demonstrated benefits, such as reducing diagnostic uncertainty, improving patient management, and improving first-attempt procedural success ratio [6–12]. Numerous medical societies endorse POCUS for these applications, with evidence linking its use to improved patient outcomes, reduced hospital stays, and lower healthcare costs [13–30].

Although initial research of its applications were mainly conducted in emergency and intensive care settings [31–34], POCUS is gaining interest in internal medicine, particularly among hospitalists [35–54]. Aligning POCUS to classical bedside evaluation has the potential to enhance the assessment of pleural effusion [55–57], pneumothorax [55–57], reduced ejection fraction and heart failure [55,56,58], deep vein thrombosis [59], and volume status [60,61]. However, research on its clinical application in internal medicine remains limited [62–66]. Its impact on key outcomes such as length of stay, readmission rates, and mortality in general inpatient settings remains underexplored [64,67–69].

As evidence indicates POCUS affects diagnostic accuracy, it is reasonable to hypothesize it may influence clinical reasoning and decision-making by refining diagnostic probabilities and reducing uncertainty in challenging cases. Although studies suggest that POCUS leads to unexpected diagnoses and management changes among hospitalists, its direct influence on clinical reasoning remains understudied [64,70–77].

This study aims to evaluate the influence of POCUS on hospitalists' clinical reasoning and decision-making by analyzing changes in diagnostic hypotheses, management strategies, and physician confidence levels.

## Materials and methods

### Study design

This is a descriptive prospective study. Research protocol was approved by our Research Ethics Committee [approval number: 6.292.577; CAAE

73039523.3.0000.0068]. Written informed consent was obtained from all participants. Data collection was conducted between October 11, 2024, and January 21, 2025, at the Transitional Care Unit of the *Instituto Perdizes* (Perdizes Institute/IPer). IPer is an institute of the Hospital das Clínicas, Faculty of Medicine, University of São Paulo, a hospital complex with 2,590 beds. The Transitional Care Unit at IPer has 70 beds and serves adult patients who require a transition from acute care to post-acute care. Its primary goals are to enhance patients' rehabilitation, educate caregivers on complex care needs, and ensure a safe transition to outpatient care. The medical team consists solely of hospitalists specialized in general internal medicine, with no on-site subspecialists available. Imaging resources are limited to conventional radiography, and advanced imaging requires inter-hospital transfer. The average patient age admitted to IPer is 58 years old. The most frequent reasons for admission are stroke, traumatic brain injury, spinal cord injury, surgical site infections, infective endocarditis, post-valve replacement surgery, and heart failure.

## Participants and procedures

All hospitalists were invited to participate. After signing the written Informed Consent Form, participants completed a form collecting personal data (Fig 1), including age, gender, education, previous POCUS experience, and any prior formal POCUS training (S1 Appendix A).

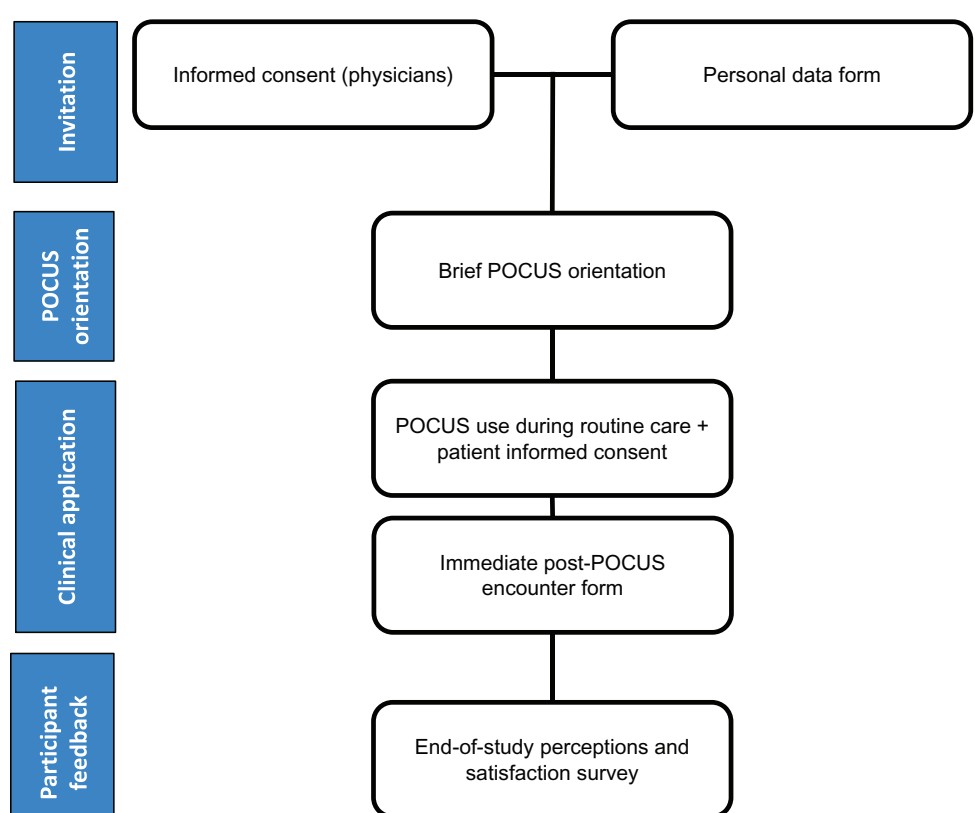

**Fig 1. Study procedures flowchart.** Hospitalists who agreed to participate signed an informed consent form (ICF) and completed a personal data form. They then attended a brief POCUS orientation. During routine care, POCUS was used at clinicians' discretion. Immediately after each use, participants completed a structured encounter form documenting the clinical question, pre- and post-POCUS diagnostic hypotheses, confidence levels, and management decisions. At the end of the study, participants completed a perceptions/satisfaction survey.

Following the initial assessment, all participants underwent an introductory educational session on POCUS (Fig 1), which covered basic applications in clinical practice, equipment use and maintenance, transducer types, image optimization, and interpretation of images from lungs, pleura, heart, abdomen, and vessels. This session was not based on a standardized protocol; rather, it provided a broad overview of commonly relevant organ systems to internal medicine hospitalists. The emphasis was on introducing participants to the potential clinical applications of POCUS rather than on training them to follow a fixed, structured protocol.

Throughout the study, participants had access to a portable ultrasound machine (Mindray DC-80A/2022) and used it as needed in clinical practice. Examinations were performed at the discretion of each physician according to the clinical question, without following a predefined or standardized protocol. Written Informed Consent was obtained from all patients who underwent POCUS examinations. Immediately after each use, participants completed a detailed form (Fig 1) documenting the reason for using POCUS, the initial diagnostic hypothesis, planned actions, the ultrasound modalities used—chosen from seven options (cardiac, pulmonary, abdominal, inferior vena cava, femoral and popliteal veins, bladder, and other), with the possibility of selecting multiple modalities per form—, the estimated duration of the exam (self-reported), the perceived quality of the images obtained (self-reported), findings, final diagnostic hypothesis, and subsequent clinical decisions (S2 Appendix B). The form was collaboratively developed by the authors, based on previous studies evaluating the diagnostic and clinical influence of POCUS in internal medicine and hospitalist settings [64,72,75,78–80], and adapted to the specific objectives of this study. However, the instrument was not formally validated, and no systematic psychometric evaluation was performed. In addition, because POCUS images were not recorded, independent verification of diagnostic accuracy, image quality, and related clinical decisions was not feasible.

Upon study completion, participants filled out a survey (Fig 1) assessing their satisfaction with POCUS training, its usefulness and relevance in clinical practice, confidence in performing POCUS, and any equipment-related challenges (S3 Appendix C).

## Sample size

The sample size was calculated using the single-proportion formula for binary outcomes [81]. The primary outcome was defined as a change in the main diagnostic hypothesis and/or management plan—both coded as mutually exclusive categories (change vs. no change). Assumptions included a 95% confidence level, a 10 percentage-point margin of error, and an expected proportion of 20% based on prior studies [64,72,75,78,79], yielding a minimum of 62 forms (S2 Appendix B). Because multiple forms were submitted per physician, observations were treated as clustered. Accordingly, we applied a design-effect adjustment using an average cluster size of 5.5 forms per hospitalist (as observed in our data, presented below) and a conservative intraclass correlation coefficient of 0.10, resulting in an adjusted requirement of approximately 90 forms (S2 Appendix B).

## Statistical analysis

Statistical analyses were performed using SigmaStat software (version 11.0) [82]. Likert scale responses, treated as ordinal categorical variables, were regrouped for simplicity: responses 1 and 2 were combined into value 1, response 3 was recoded as value 2, and responses 4 and 5 were grouped into value 3. The following variables were recoded according to this categorization approach: confidence in performing POCUS (S1 Appendix A); diagnostic confidence before and after POCUS and image quality (S2 Appendix B); as well satisfaction, perceived usefulness, and importance of POCUS training; confidence in equipment use and image interpretation; and interest in future use of POCUS (S3 Appendix C).

Descriptive statistics were used to summarize the characteristics of the study participants and the collected data. Categorical variables were expressed as absolute and relative frequencies. To assess the distribution of continuous variables, the Shapiro-Wilk test was performed. Since none of the variables followed a normal distribution, continuous variables are presented as medians and the 25th and 75th percentiles. Spearman rank correlation was performed to explore

associations between continuous and ordinal variables. A significance level of $p < 0.05$ was considered statistically significant for all analyses.

In addition, open-ended question responses and relevant closed-ended items from the POCUS clinical-use form (S2 Appendix B) were transcribed into a structured Excel sheet and coded by the principal investigator for each form. Open-ended items used in the analyses were: primary diagnostic hypothesis (pre/post-POCUS), differential diagnoses (pre/post-POCUS), planned management (pre/post-POCUS), expected findings, and POCUS findings. The closed-ended item used was the diagnostic confidence in the primary hypothesis (pre/post-POCUS). From these fields, we created binary indicators (yes/no) for: change in the primary diagnostic hypothesis; change in diagnostic confidence; addition of an alternative diagnosis; exclusion of an alternative diagnosis; incidental findings; and change in management. Change in diagnostic confidence was defined as a shift between the three recoded Likert categories (value 1 = Likert 1–2; value 2 = Likert 3; value 3 = Likert 4–5). These derived variables were summarized descriptively as categorical data, and no formal qualitative analysis was performed. Open-text responses were also reviewed to extract brief contextual information.

## Results

### Participants' characteristics

Of the 26 hospitalists invited, 19 were enrolled. Participant demographics and characteristics are summarized in Table 1.

Spearman rank correlation analysis revealed a significant negative correlation was found between the time since residency completion and having formal POCUS training ($R = -0.479$; $p = 0.0374$). While a positive correlation trend was observed between the frequency of POCUS use prior to the study and participants' self-confidence ($R = 0.431$; $p = 0.0649$), it did not reach statistical significance.

**Table 1. Participants' characteristics.**

| Characteristics | Results (N = 19) |
|---|---|
| Median age – yr (IQR)[a] | 29 (28.0; 31.0) |
| Gender | |
| Male – no. (%) | 9 (47) |
| Female – no. (%) | 10 (53) |
| Nationality | |
| Brazilian – no. (%) | 19 (100) |
| Undergraduate degree besides medicine | 0 |
| Specialization | |
| Internal Medicine only – no. (%) | 15 (79) |
| Additional specialization – no. (%) | 4 (21) |
| Time passed since residency training – yr (IQR) [a] | 3.0 (2.0; 3.75) |
| Formal training on POCUS | |
| Yes – no. (%) | 5 (26) |
| POCUS usage prior to study – no. of times (IQR) [a] | 15.0 (5.0; 30.0) |
| Self-confidence in POCUS – no. (%) | |
| Slightly or not at all confident | 6 (31) |
| Confident | 7 (37) |
| Very or extremely confident | 6 (31) |

The table presents demographic data and key characteristics, including age, gender distribution, nationality, academic background, and POCUS-related information.

[a]Interquartile ranges.

## Satisfaction and perspectives of the application of POCUS

Fifty-three per cent (10/19) of participants completed the survey regarding training satisfaction and personal perspectives of the application of POCUS. All respondents were either "very " or "extremely satisfied" with the training and agreed that it improved their ultrasound skills. Participants also highly valued POCUS, with all considering it "very" or "extremely useful" in clinical practice, and they all agreed that formal POCUS training is essential in medical residency programs. Additionally, everyone expressed strong interest in using POCUS in their future practice.

Regarding self-confidence while using POCUS on clinical scenarios, 60% (6/10) felt "confident," 30% (3/10) felt "very" or "extremely confident," and 10% (1/10) felt "slightly" or "not at all confident." No correlation was observed between previous POCUS training and self-confidence levels (R = 0.150; p = 0.531). Participants reported challenges with ultrasound handling (4/10), image interpretation (5/10), need for more hands-on POCUS training for independent use (1/10), and limited time for using POCUS in clinical settings (3/10).

## Clinical applications of POCUS

In total, 104 forms on POCUS clinical use were completed by 19 participants throughout the study, with a median of 2.0 forms [1.00; 5.75] per participant. Each form assessed one clinical encounter, which could demand more than one POCUS modality and clinical purposes. Among the 104 forms, POCUS was used by participants in various modalities and for multiple clinical purposes, as summarized in Figs 2 and 3, respectively.

The total time spent using POCUS was 935 minutes, with a median value of 10 minutes per patient [5.00; 10.00].

Regarding image quality, participants showed varying levels of satisfaction: 1/104 form(0.9%) was "slightly satisfied," 21/104 forms (20%) were "satisfied," and 82/104 forms (79%) were "very" or "extremely satisfied".

## Influence of POCUS on clinical reasoning and decision-making

Out of the 104 times POCUS were performed on patients, there was a change in the primary hypothesis in 38/104 scenarios (37%), while the primary diagnostic hypothesis remained unchanged in 66/104 scenarios (63%) (Fig 4). In 41/104

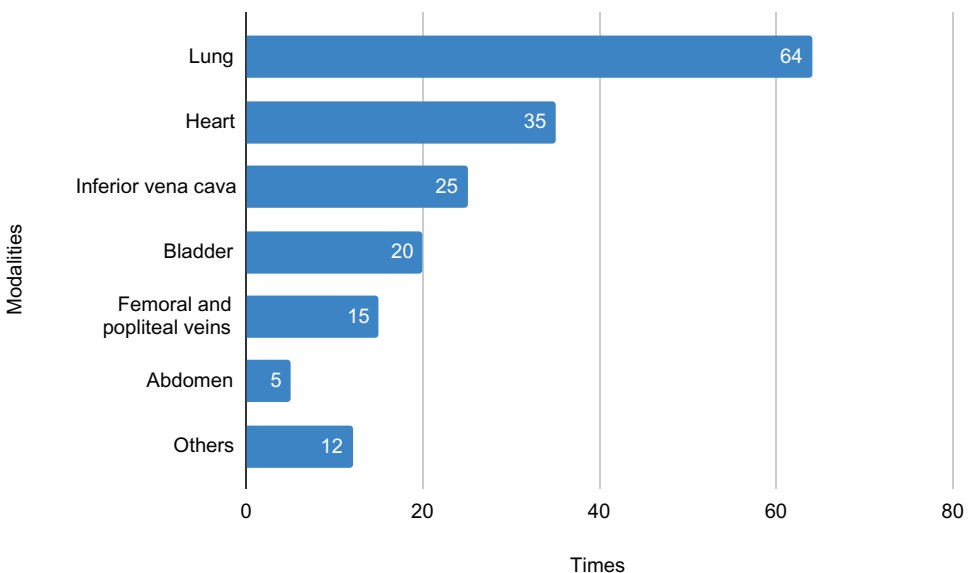

**Fig 2. POCUS modalities used.** The figure shows the distribution of POCUS use across various modalities, with lung being the most frequent (64 times), followed by heart (35 times) and inferior vena cava (25 times). More than one POCUS modality could be used in the same patient.

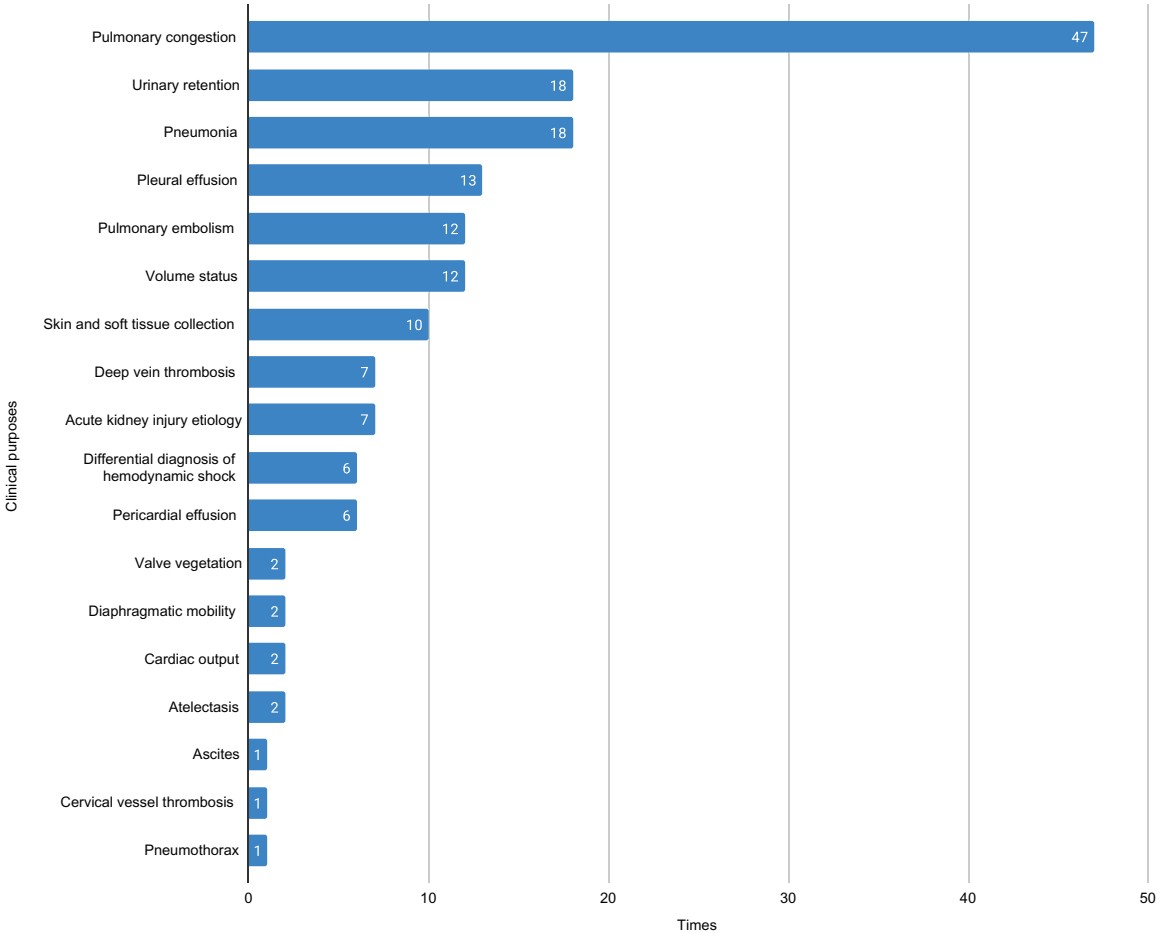

**Fig 3. Clinical purposes.** The figure illustrates the number of times POCUS was employed for different diagnoses, with the most frequent uses for pulmonary congestion (47 times), urinary retention and pneumonia (18 times each) and pleural effusion (13 times).

(39%) forms, there was a change in management, while in 63/104 (60%) management was maintained (Fig 5). In 48/104 forms (46%), there was an increase in diagnostic confidence, while in 56/104 (54%), confidence remained unchanged or decreased (Fig 4). Of the 104 completed forms, only 14/104 (13%) showed no influence of POCUS on clinical reasoning or decision-making. To assess this influence, we considered the following criteria: changes in the primary diagnostic hypothesis, changes in confidence level regarding the primary hypothesis, inclusion of a diagnostic hypothesis, exclusion of a hypothesis, identification of incidental findings, and modifications in clinical management.

Among the 66 cases where the primary hypothesis was maintained, there was an increased level of confidence in 24/66 (36%). Twenty-two shifted from "confident" to "very confident" or "extremely confident," and 2 shifted from "slightly confident" to "very confident." No difference in the level of confidence was observed in 41/66 (62%). Amid cases with no confidence changes, 39/41 (95%) already had "high" or "extreme" confidence before performing POCUS, and the findings in these cases were consistent with expectations. The participants informed a decrease in their level of confidence in 1/66 (2%) scenario (Fig 4), shifting from "very confident" to "slightly confident".

In 41/66 (62%) of the cases, participants used POCUS to rule out a diagnostic hypothesis, while 4/66 (6%) yielded incidental findings (Table 2). Only 1/66 (2%) case resulted in the addition of a diagnostic hypothesis based on POCUS findings (Table 2). Despite maintaining the primary diagnostic hypothesis, clinical management was altered in 5/66 (8%)

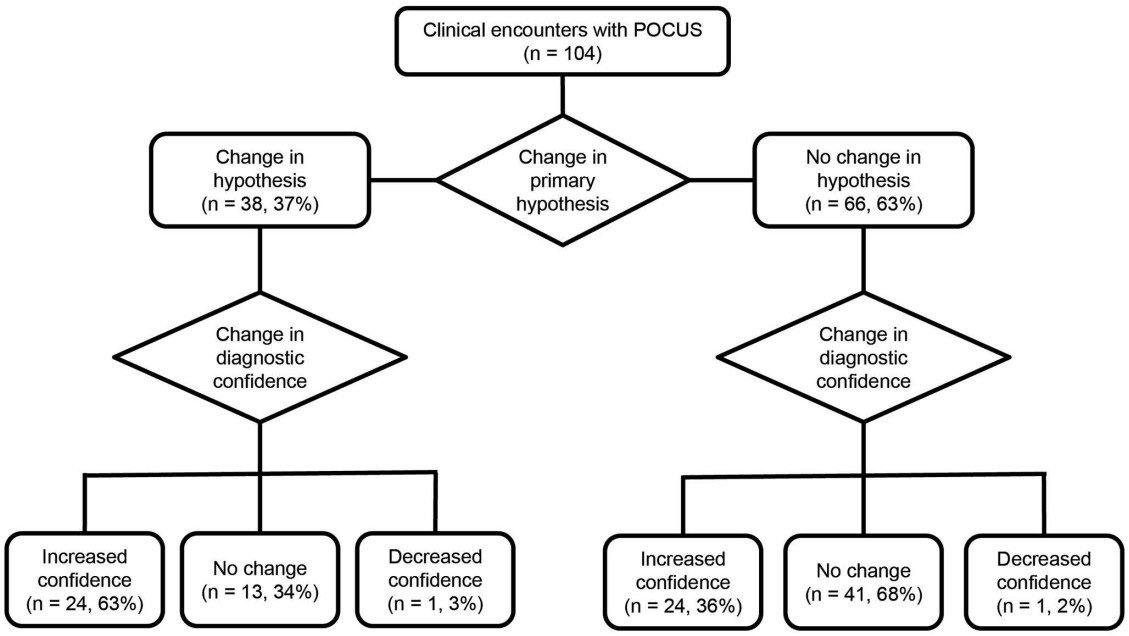

**Fig 4. Diagnostic influence.** The flowchart illustrates that in 37% of cases (38/104), the primary diagnostic hypothesis was modified following POCUS, whereas in 63% (66/104), it remained unchanged. Among cases with no change in hypothesis, diagnostic confidence increased in 36% (24/66), remained the same in 62% (41/66), and decreased in 2% (1/66). In cases where the hypothesis changed, 66% (25/38) showed a change in diagnostic confidence—96% of these (24/25) reported increased confidence, while 4% (1/25) reported decreased confidence. The remaining 34% (13/38) reported no change in confidence.

cases (Fig 5). In 3 of these, POCUS was used to assess pulmonary congestion, and although findings were consistent with expectations, participants compared them with prior exams and determined that congestion was improving, opting not to intensify diuretic therapy. In 1 case, management was adjusted after ruling out an alternative hypothesis, and in another, an angiotomography was added, as POCUS was not considered sufficient to exclude pulmonary thromboembolism.

Of the 38 forms where the primary diagnostic hypothesis changed, only 2/38 (5%) did not result in a change in management (Fig 5), as the initial planned strategy included both the pre-POCUS and post-POCUS primary diagnostic hypotheses. In 13/38 (34%) of these forms, the level of confidence in the primary diagnostic hypothesis remained unchanged (Fig 4). Among these, 2 participants were "confident," 10 were "very confident" or "extremely confident" and 1 was "slightly confident" regarding the diagnosis. In the other 25/38 (66%) that altered their level of confidence in the primary diagnostic hypothesis, only 1/25 (4%) showed a decrease (Fig 4), shifting from "confident" to "slightly confident". The remaining 24/25 (96%) recorded increases: 16 moved from "confident" to "very" or "extremely confident", 5 from "slightly confident" to "very" or "extremely confident" ", 1 from "slightly confident" to "confident," 2 from "not confident" to "very" or "extremely confident".

In 3/38 forms (8%), incidental findings were recorded (Table 2); however, in only 1 case the finding was significant enough to add a new diagnostic hypothesis, which became the primary one, and resulted in changes in clinical management. In 32/38 forms (84%), a diagnostic hypothesis was ruled out (Table 2): 30 ruled out the primary diagnostic hypothesis, 1 ruled out an alternative diagnostic hypothesis, and 1 identified a more likely alternative diagnosis, replacing the primary hypothesis with this alternative diagnosis. In all cases where the primary diagnostic hypothesis changed, the POCUS findings differed from the participants' initial expectations.

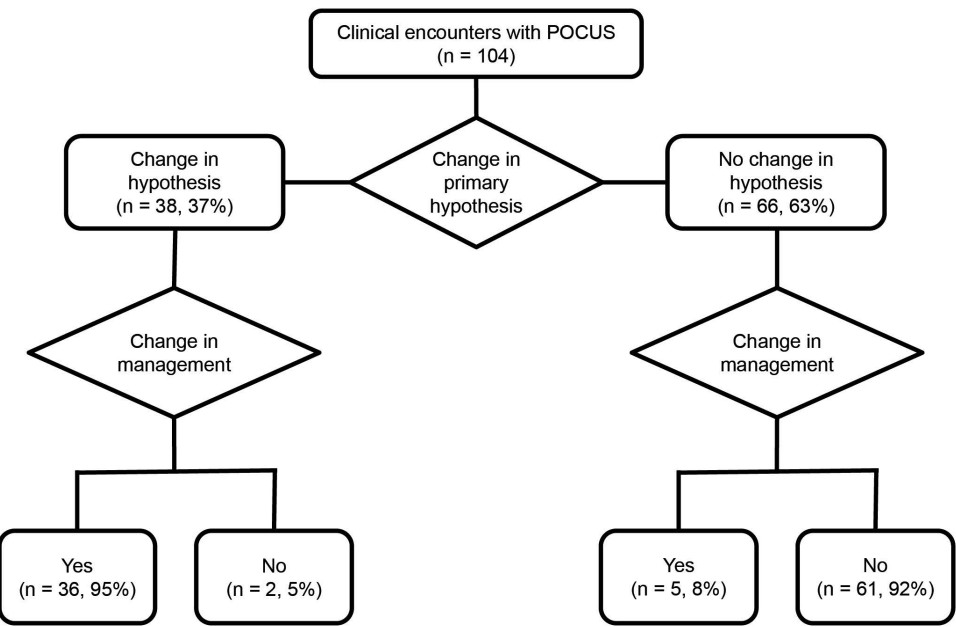

**Fig 5. Clinical decision-making.** The flowchart illustrates that when the primary diagnostic hypothesis changed (38/104 cases), a change in clinical management occurred in 95% of those cases (36/38). When the hypothesis remained unchanged (66/104 cases), a change in management was observed in 8% of cases (5/66).

**Table 2. POCUS influence on diagnostic reasoning and decision-making when the primary diagnostic hypothesis was maintained vs. changed.**

| Outcome | Primary hypothesis maintained (N = 66) | Primary hypothesis changed (N = 38) |
|---|---|---|
| **Increased confidence** | 24 (36%) | 24 (63%) |
| **Ruling out hypothesis** | 41(62%) | 32 (84%) |
| **Incidental findings** | 4 (6%) | 3 (8%) |
| **Addition of hypothesis** | 1 (2%) | 0 (0%) |
| **Change in management** | 5 (8%) | 36 (95%) |

Distribution of diagnostic confidence, exclusion of diagnostic hypotheses, incidental findings, addition of diagnostic hypotheses, and management decisions, according to whether the primary hypothesis was maintained or changed.

## Discussion

Our study suggests that POCUS may influence clinical reasoning and decision-making. In 37% (38/104) cases, POCUS led to a modification of the primary diagnostic hypothesis, while in 39% (41/104), it resulted in changes to the clinical management plan. Additionally, an increase in diagnostic confidence was observed in 46% (48/104) of cases. Notably, in only 13% (14/104) of instances, POCUS had no discernible effect on clinical reasoning or decision-making.

These findings align with Lucas et al. [64], who reported management changes in 37% of the 210 participants who underwent cardiac ultrasound and conventional echocardiogram, but show a greater impact compared to Smallwood et al. [78], where only 34.8% of the 276 participants reported weekly decision-making changes and 10.3% daily. Compared to Cid-Serra et al. [72], who conducted a systematic review encompassing six studies with a total of 1,836 patients, our

study found a higher rate of primary diagnostic hypothesis changes (37% vs. 18%), though clinical management influence was similar (39% vs. 37–52.1%). Similarly, our results exceeded those of Andersen et al. [79], who assessed 199 patients examined by medical residents. They reported changes in the primary diagnosis in 6.5% of cases. Our findings also indicate a greater diagnostic influence than Mjolstad et al. [75], who studied the diagnostic effect of cardiac and abdominal screening in 196 hospitalized patients and found a primary diagnosis change in 18.4% of cases.

In our study, even when the primary hypothesis remained unchanged (63% − 66/104), POCUS played an important role in strengthening diagnostic confidence, with certainty increasing in 36% (24/66) of these cases. The absence of changes in diagnostic confidence in some forms can be explained by the ceiling effect [83–86]. Many participants who showed no variation in confidence were already classified as "very" or "extremely" confident in their initial diagnostic hypothesis before performing POCUS. In these cases, the ultrasound served primarily as a confirmatory tool, with little room for a significant increase in confidence. This phenomenon is expected in studies assessing confidence variation, as participants who start at a maximum level of certainty are less likely to demonstrate additional impact (ceiling effect), even when the tool used is effective.

POCUS also played an essential role in narrowing differential diagnoses, a particularly valuable function for hospitalists managing complex cases. When the primary hypothesis was maintained, POCUS helped rule out diagnoses in 62% (41/66) of cases, streamlining clinical decision-making. While management remained unchanged in most instances, POCUS still influenced care by guiding therapeutic choices in 8% (5/66), often through trend assessments or the exclusion of competing diagnoses. These findings highlight how POCUS not only supports diagnostic confirmation but also helps hospitalists refine differential diagnoses, potentially contributing to patient management decisions.

When the primary diagnostic hypothesis changed, POCUS had an even greater influence. In 95% (36/38) of these cases, management was modified, suggesting its direct influence on patient care. Additionally, in 84% (32/38) of cases, a diagnosis was ruled out, highlighting POCUS's role in refining differential diagnoses—a key advantage for hospitalists navigating complex decision-making. Confidence levels also shifted notably, with 63% (24/38) of cases showing an increase in diagnostic certainty, further emphasizing POCUS's role in strengthening clinical confidence.

Incidental findings were rare (6.7% − 7/104). Among these, only one case led to a complete change in the primary diagnosis, and another added an alternative diagnostic hypothesis. This rarity likely reflects that clinical reasoning is primarily guided by history-taking and physical examination, which help establish an initial diagnostic framework and set expectations for POCUS findings. Consequently, POCUS in our setting mainly served to confirm or refine initial hypotheses rather than introduce unexpected diagnoses. However, the identification of incidental findings also raises an important consideration. Such findings may divert the use of POCUS from its primary purpose—providing rapid answers to focused, binary clinical questions at the bedside—toward a more exploratory or comprehensive approach that it is not designed to fulfill. This highlights the need to maintain a clear diagnostic focus and to apply POCUS judiciously, ensuring it enhances rather than detracts from clinical reasoning. Notably, its role is particularly relevant in cases with intermediate pretest probability, where additional imaging can meaningfully influence decision-making by increasing certainty or ruling out alternative diagnoses.

Although limited data exist on the risk of incidentalomas in point-of-care ultrasound, there is growing concern in the literature about this issue [87–89]. Unfocused or overly broad imaging may lead to incidental findings that trigger unnecessary investigations, patient anxiety, and resource use. However, when POCUS is guided by a specific clinical question and integrated into a structured diagnostic process, the chance of encountering incidentalomas is greatly reduced. This underscores the importance of training clinicians to use POCUS in a hypothesis-driven manner within the context of clinical reasoning. Moreover, healthcare teams and institutions should anticipate the possibility of incidental findings and develop protocols to manage them appropriately.

One of the key advantages of POCUS is its utility in settings with limited access to traditional radiologic imaging, such as the Transitional Care Unit at IPer. This unit has only plain radiography available, requiring inter-hospital transfer for specialized evaluations or additional imaging. This is also important in rural practices, humanitarian missions, and conflict

zones [90–93]. In these scenarios, POCUS might serve as an immediate diagnostic tool, enabling physicians to obtain crucial bedside information, optimizing clinical decision-making and management. Furthermore, given that the Transitional Care Unit at IPer primarily cares for elderly patients with multiple comorbidities and complex conditions, such as stroke, traumatic brain injury, and heart failure, the use of POCUS may help reduce reliance on additional imaging, minimize unnecessary transfers, and enhance overall care efficiency, ultimately facilitating a safe hospital discharge.

POCUS was applied across diverse clinical contexts, including cardiopulmonary, urinary, and hemodynamic assessments, underscoring its versatility in hospital-based practice. While our study did not evaluate the impact of a systematic approach, the breadth of its use observed is consistent with existing literature that supports structured protocols to improve the consistency and diagnostic accuracy of POCUS [94–99]. Such approaches may enhance bedside clinical decision-making, particularly in acute care settings.

The participants were a homogeneous group in terms of academic background and clinical experience, consisting mostly of young professionals early in their careers. They had limited formal POCUS training, and their practical experience varied, leading to differences in confidence. These findings align with previous studies show that most Internal Medicine physicians use POCUS despite limited training [78,100].

Beyond the direct findings, our study highlights broader implications for hospital medicine. First, the frequent use of POCUS by hospitalists in diverse scenarios suggests that it is becoming integrated into routine reasoning processes. By influencing diagnostic hypotheses, management strategies, and confidence levels, POCUS appears to shape clinical reasoning in meaningful ways. However, most participants reported limited prior formal training, reflecting the well-documented gap in structured POCUS education within internal medicine residency programs [78,100]. This gap raises important concerns. The use of POCUS without adequate training may increase the risk of misinterpretation and diagnostic error, undermining the tool's potential benefits. Previous studies have emphasized that appropriate supervision, competency standards, and validated curricula are essential to ensure safe and effective integration of POCUS into practice [78,100–103]. Therefore, our findings reinforce the need for internal medicine societies and residency program directors to systematically consider how POCUS training should be incorporated into education and clinical governance frameworks.

Furthermore, the survey results revealed a highly positive perception of brief POCUS orientation session. All participants expressed satisfaction, emphasizing its value in clinical practice and unanimously supporting its inclusion in residency programs and continuing medical education programs. However, barriers such as ultrasound handling, image interpretation, limited supervision, and clinical time constraints were noted.

Although no significant correlations were found between prior formal training and self-confidence, it is important to highlight that self-confidence in using POCUS in clinical scenarios was assessed in only 10 out of 19 respondents (53%), which limits the strength of conclusions regarding confidence and should be considered a limitation.

This study has some limitations. First, we did not assess the impact of POCUS on robust clinical outcomes such as length of stay, readmission rates, or mortality. Second, the small sample size may limit the generalizability of the results and increase susceptibility to bias. Third, POCUS images were not recorded, precluding independent review of acquisition and interpretation, and preventing evaluation of diagnostic accuracy, interobserver reliability, or concordance with gold-standard criteria. Decisions were therefore based solely on participants' self-reported findings, which may have led to overestimation of perceived safety and influence. In addition, process variables such as exam duration and image quality were also self-reported rather than objectively measured. Exam time was estimated by participants, raising the risk of recall bias, while image quality was judged subjectively without external validation, introducing self-report bias. Another limitation relates to the data collection form, which was developed by the authors based on previous studies and study objectives but was not formally validated; as such, the accuracy and interpretability of the captured constructs may be constrained. Finally, open-ended responses were categorized by a single investigator without independent coding or interrater reliability assessment, which may introduce subjective bias. These aspects should be considered when interpreting the feasibility and quality of POCUS implementation in our study.

Despite these limitations, our findings underscore the meaningful influence of POCUS on the clinical practice of hospitalists, highlighting its important role in refining clinical reasoning and decision-making. The findings indicate that POCUS contributed to a change in the primary diagnostic hypothesis in 37% (38/104) of cases, a modification in clinical management in 39% (41/104), and an increase in diagnostic confidence in 46% (48/104) of cases. Additionally, the use of POCUS in ruling out differential diagnoses may contribute to refining clinical reasoning, which is often complex in hospital medicine. Moreover, POCUS has potential to assist hospitalists in settings with limited access to imaging exams, where bedside ultrasound serves as a useful adjunct to support timely decision-making, potentially guiding patient care and management.

## Conclusion

This study suggests that POCUS may influence diagnostic hypotheses, clinical management decisions, and physician confidence. It appeared to contribute to clinical reasoning and decision-making in many cases, while in some instances it showed little or no effect.

These findings add to the limited body of literature on the application of POCUS in internal medicine, indicating its potential value as a bedside modality. By illustrating its possible influence on clinical judgment and reasoning—particularly in resource-limited settings—this study provides preliminary evidence that POCUS may serve as a useful adjunct in hospitalist practice. Future research should employ stronger designs, such as randomized controlled trials or controlled before-and-after studies with validated outcome measures and objective performance data, to determine whether enhancements in clinical reasoning facilitated by POCUS translate into measurable improvements in patient outcomes and healthcare efficiency, such as reduced hospital length of stay, readmission rates, or mortality.

Overall, the results should be interpreted as exploratory, emphasizing feasibility and potential utility rather than efficacy. As clinical demands continue to evolve, incorporating effective bedside approaches like POCUS could ultimately play a supportive role in improving patient care.

## Supporting information

**S1 Appendix. A. Form on participants' personal data.** Questionnaire completed by participants before the study. It collects demographic data, academic background, previous experience with point-of-care ultrasound (POCUS), and self-assessed confidence in performing and interpreting POCUS examinations.
(DOCX)

**S2 Appendix. B. Form on POCUS clinical use.** Structured form used by participants after each POCUS application during clinical practice. It documents the clinical question, diagnostic hypothesis, confidence levels before and after POCUS, selected ultrasound modalities, image quality, findings, and subsequent changes in diagnosis or management.
(DOCX)

**S3 Appendix. C. Form on participants' impressions and satisfaction survey.** Survey assessing participants' satisfaction with the POCUS course, perceived usefulness in clinical practice, importance of formal training, confidence levels after training, future interest in using POCUS, and perceived barriers to its implementation in the clinical setting.
(DOCX)

## Acknowledgments

We would like to thank Instituto Perdizes for their invaluable support in conducting this research.

## Author contributions

**Data curation:** Maira Dias Souza, Hassan Rahhal, Iolanda F. L. C. Tibério.

**Formal analysis:** Maira Dias Souza, Hassan Rahhal, Iolanda F. L. C. Tibério.

**Investigation:** Maira Dias Souza, Hassan Rahhal, Iolanda F. L. C. Tibério.

**Methodology:** Maira Dias Souza, Hassan Rahhal, Iolanda F. L. C. Tibério.

**Project administration:** Maira Dias Souza, Hassan Rahhal, Iolanda F. L. C. Tibério.

**Supervision:** Maira Dias Souza, Hassan Rahhal, Iolanda F. L. C. Tibério.

**Validation:** Maira Dias Souza, Hassan Rahhal, Iolanda F. L. C. Tibério.

**Visualization:** Maira Dias Souza, Hassan Rahhal, Iolanda F. L. C. Tibério.

**Writing – original draft:** Maira Dias Souza, Hassan Rahhal, Iolanda F. L. C. Tibério.

**Writing – review & editing:** Maira Dias Souza, Hassan Rahhal, Iolanda F. L. C. Tibério.

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
