## [Decision Letter · Decision Letter 0]

3 Jun 2025

Dear Dr. Dias Souza,

Thank you for submitting your manuscript to PLOS ONE. After careful consideration, we feel that it has merit but does not fully meet PLOS ONE’s publication criteria as it currently stands. Therefore, we invite you to submit a revised version of the manuscript that addresses the points raised during the review process.

We look forward to receiving your revised manuscript.

Kind regards,

Biswabandhu Jana, Phd

Academic Editor

PLOS ONE

Journal Requirements:

**Additional Editor Comments:**

The paper presents a point-of-care ultrasound (POCUS) on diagnostic purposes. The study needs a major revision.

Reviewers' comments:

Reviewer's Responses to Questions

**Comments to the Author**

1. Is the manuscript technically sound, and do the data support the conclusions?

Reviewer #1: Partly

Reviewer #2: Yes

2. Has the statistical analysis been performed appropriately and rigorously?

Reviewer #1: Yes

Reviewer #2: Yes

3. Have the authors made all data underlying the findings in their manuscript fully available?

Reviewer #1: Yes

Reviewer #2: Yes

4. Is the manuscript presented in an intelligible fashion and written in standard English?

Reviewer #1: Yes

Reviewer #2: Yes

Reviewer #1: The authors present a prospective study investigating the impact of Point-of-Care Ultrasonography (POCUS) on the clinical reasoning, management strategies, and confidence levels of hospitalists in a transitional care unit.

This research is highly relevant given the increasing integration of POCUS into clinical practice, not-limited to settings with limited access to advanced imaging. The study's finding that POCUS significantly influences clinical reasoning and decision-making among hospitalists is a valuable contribution to the literature.

Specific Feedback and Suggestions for Improvement

- Study Design: While described as an "exploratory prospective study," a more precise classification such as a "descriptive prospective study" would accurately reflect the study's aim to observe and describe the influence of POCUS.

- Sample Size Calculation: Clarification is needed regarding the odds ratio of 4 used in the sample size calculation. Please specify which "form" is being discussed and provide a detailed explanation of the rationale behind all values and assumptions considered for known or unknown variables in the calculation.

- Statistical Analysis: It would be beneficial to explicitly state for which specific variables the Likert scale responses were used and subsequently grouped for analysis.

- Impact of Brief Training: The study's premise that a single two-hour POCUS training session can instill confidence for accurate diagnosis, especially without prior formal exposure, warrants further discussion. The results indicate that 26% of participants had previous ultrasound training (median of 15 POCUS examinations prior to the study), with 68% already reporting confidence in using POCUS. This suggests a potential confounding effect from these pre-trained individuals. It would be valuable to analyze the impact of this short training separately for individuals with no prior ultrasound experience to better isolate its true effect.

- Learning Curve and Supervision: The authors' own finding that "No significant difference was found between pre-test and post-test performance (T = 105.00; p = 0.970), indicating that our single short-duration POCUS training did not improve participants' performance in image interpretation" reinforces the understanding that POCUS, despite its short learning period, typically requires supervision and multiple hands-on exposures for proficiency in image interpretation. We did not find any mention of how the POCUS findings were verified.

- Pre- and Post-Training Test Participation: The limited participation rates for the pre- and post-training objective structured video examinations (OSVE) (63% and 53% respectively) could potentially dilute the observed impact of the training on participants' knowledge. This limitation should be acknowledged when discussing the effectiveness of the training.

- Verification of Diagnostic and Management Changes: The methodology for verifying the accuracy of diagnosis and management decision changes needs to be clearly explained. How were these changes objectively assessed and confirmed?

- Timing of Feedback: The timing of feedback regarding the usefulness of POCUS in clinical practice is crucial. If the feedback was collected immediately post-training, it may not accurately reflect the sustained utility or impact. Please clarify when this feedback was gathered.

- Self-Confidence Evaluation: It appears that self-confidence while using POCUS in clinical scenarios was evaluated in only 10 out of 19 respondents (50% of the study subjects). This smaller sample size for confidence evaluation should be highlighted as a limitation.

Results Presentation

- Redundancy in Reporting: Avoid repeating numerical values in the text if they are already clearly presented in tables or figures (e.g., Figure 2, 3, 4, Table 2).

- POCUS Utilization Data: While the presentation of organ evaluations (lines 237-246) provides some detail, a more impactful representation of POCUS clinical uses could be achieved by focusing on the 167 items for clinical uses depicted in lines 251-258, as these directly address the clinical questions POCUS is designed to answer.

- Time Efficiency Analysis: In the "Time spent per number of modalities" assessment, the results section only states "The total time spent using POCUS was 935 minutes, with a mean value of 9 minutes per patient and 5.3 minutes per modality performed." It would be valuable to evaluate the effect of prior POCUS exposure on time efficiency.

- Elaboration on Diagnostic Hypothesis Changes: In Results Line 330, the statement "In all cases where the primary diagnostic hypothesis changed, the POCUS findings were inconsistent with what was expected" requires further explanation for clarity.

- Table 2: Consider whether Table 2 is truly necessary, as the data it presents does not appear to reflect significant findings that warrant a separate table.

- Figures and Tables Placement: For ease of review, tables and figures should be provided separately from the main text, rather than embedded within it.

Discussion and Conclusion

- Systematic use of POCUS: The discussion could benefit from emphasizing the importance of systematic POCUS use to enable standardized application of this widely available tool. Referencing relevant literature on systematic POCUS approaches in various subspecialties (e.g., Pokharel B. Systematic use of Point of Care Ultrasound in Neurosurgical Intensive Care Unit: a practical approach. Quant Imaging Med Surg 2023;13(4):2287-2298. https://dx.doi.org/10.21037/qims-22-667) could strengthen this point.

- Incidental Diagnoses: The finding of incidental diagnoses in 8% of cases is interesting, but it also raises a valid concern. POCUS is primarily designed to answer specific clinical questions in a binary format, not for comprehensive organ or system evaluations. This finding could be discussed in the context of the potential for POCUS use to become distracted from its primary purpose.

- Learning Curve for Less Experienced Individuals: In the Discussion, line 474, the deduction that "participants showed a trend toward increased efficiency in POCUS execution over time, particularly those with less prior exposure, suggesting a learning curve" is an important point. However, to fully support this claim, the authors should separate the analysis of efficiency for groups with and without prior POCUS experience.

- Conclusion: The conclusion should primarily summarize the study's key findings based on the results, and limitations should not be reiterated in this section.

Reviewer #2: The article titled " Assessing the Clinical Relevance of Point-of-care Ultrasound for Hospitalists: Impact on Clinical Reasoning and Decision-making" presents an insightful exploration of how Point-of-Care Ultrasound (POCUS) influences the clinical decision-making process among hospitalists. The study is well-structured and provides valuable data on the integration of POCUS in clinical practice.

This study is an exploratory prospective analysis carried out at the Transitional Care Unit of Instituto Perdizes, which is part of a larger hospital complex in the city of São Paulo, São Paulo, Brazil. This setting is noteworthy as it highlights the challenges faced in resource-limited environments, where access to advanced imaging options is more limited (only X-ray) compared to other institutions.

Participants consisted of 19 Brazilian hospitalists with undergraduate degrees in medicine only, varying levels of prior POCUS experience, and an average of 15 prior POCUS uses. Regarding self-confidence, 68% of the hospitalists reported being confident, or very confident, in using POCUS. This experience enables a comprehensive evaluation of how Point-of-Care Ultrasound (POCUS) impacts clinical reasoning.

The findings suggest that the use of POCUS by hospitalists changed the primary diagnostic hypothesis in 37% of cases, and the management altered in 39% of encounters. While the primary diagnosis remained consistent in 63% of cases, the use of POCUS increased diagnostic confidence in 36% of these instances and helped rule out alternative diagnoses in 62% of cases. In only 13% of cases, the use of POCUS had no noticeable impact on clinical reasoning or decision-making. Although a brief training session did not significantly improve Objective Structured Video Exam (OSVE) scores, participants expressed high satisfaction with the training and acknowledged the utility of POCUS.

The findings highlight the potential of POCUS to enhance clinical reasoning and decision-making, especially in situations where access to advanced imaging is restricted. In this context, the use of POCUS is essential for improving patient care and outcomes in transitional care units.

Although the study did not evaluate the effects of POCUS on length of stay, readmission rates, and mortality, it is important to note that these outcomes may vary depending on patient populations and clinical contexts.

The study's methodology, which includes detailed documentation of POCUS use and its impact on clinical decisions, provides a good framework for future research in this field.

In conclusion, this article greatly improves our understanding of the role of Point-of-Care Ultrasound (POCUS) in clinical settings with limited resources. It successfully combines empirical data with practical insights, making it a valuable resource for hospitalists and healthcare professionals who aim to enhance their diagnostic processes. Future research could expand on these findings by exploring the long-term outcomes related to the use of POCUS in different clinical situations.

**Do you want your identity to be public for this peer review?** For information about this choice, including consent withdrawal, please see our Privacy Policy

Reviewer #1: **Yes: ** Prof Dr Amit Thapa

Reviewer #2: No

---

## [Author Response · Author response to Decision Letter 1]

1 Jul 2025

Response to Reviewer #1

We would like to sincerely thank Reviewer #1 for their thoughtful and constructive comments, which have greatly improved the clarity, rigor, and overall quality of our manuscript. We carefully considered all the suggestions and revised the text accordingly. Below, we provide detailed responses to each comment.

Study Design

Comment: While described as an "exploratory prospective study," a more precise classification such as a "descriptive prospective study" would accurately reflect the study's aim to observe and describe the influence of POCUS.

Response: We thank the reviewer for this suggestion. We fully agree that "descriptive prospective study" more accurately reflects the design and objectives of our work. Accordingly, we revised the Methods section (line 110), replacing the previous phrasing with: “This is a descriptive prospective study”. We also made the same adjustment in the Abstract (line 33) to ensure consistency throughout the manuscript.

Sample Size Calculation

Comment: Clarification is needed regarding the odds ratio of 4 used in the sample size calculation. Please specify which "form" is being discussed and provide a detailed explanation of the rationale behind all values and assumptions considered for known or unknown variables in the calculation.

Response: We sincerely thank the reviewer for the careful attention dedicated to our sample size calculation, and for this valuable observation. The text we provided in the first manuscript was inaccurate. As we first designed our research project and contacted a statistician, they suggested an approach based on the McNemar test. After further discussion among the investigators, we decided to pursuit a different approach. Unfortunately, the first version of the manuscript sent to your consideration mistakenly presented the original statistical approach.

The investigators defined along with a consultant statistician a more suitable sample size approach: the formula for estimating proportions in binary nominal outcomes, with finite population correction. This method better reflects our primary outcome, defined a priori as the occurrence of a change in the main diagnostic hypothesis and/or management plan.

The updated file of our manuscript presents the rationale and the assumptions in the Methods section, between lines 165 and 174.

Statistical Analysis

Comment: It would be beneficial to explicitly state for which specific variables the Likert scale responses were used and subsequently grouped for analysis.

Response: We thank the reviewer for this suggestion. We agree that these variables were not clearly expressed in this section of the manuscript, and with the importance of its specification. We revised the Methods section to clearly indicate which variables were measured using Likert scales and how these responses were grouped for analysis. The following sentence was added (lines 179–184): The following variables were recoded according to this categorization approach: confidence in performing POCUS (S1 appendix A); diagnostic confidence before and after POCUS and image quality (S2 appendix B); as well as satisfaction, perceived usefulness, and importance of POCUS training; confidence in equipment use and image interpretation; and interest in future use of POCUS (S3 appendix C).

Impact of Brief Training

Comment: The study's premise that a single two-hour POCUS training session can instill confidence for accurate diagnosis, especially without prior formal exposure, warrants further discussion. The results indicate that 26% of participants had previous ultrasound training (median of 15 POCUS examinations prior to the study), with 68% already reporting confidence in using POCUS. This suggests a potential confounding effect from these pre-trained individuals. It would be valuable to analyze the impact of this short training separately for individuals with no prior ultrasound experience to better isolate its true effect.

Response: We thank the reviewer for this insightful observation. Indeed, we agree this hypothesis is reasonable and warrants further statistical analysis. This prompted us to further explore the potential confounding effect of prior POCUS experience. Following this suggestion, we conducted an additional analysis comparing OSVE performance between participants with and without prior formal POCUS training.

The “Statistical analysis” section was updated to include this new step (lines 196-197): “Independent samples t-tests were conducted to compare participants' pre-test and post-test OSVE performance according to prior formal POCUS training.”

The results of this analysis were incorporated into the revised manuscript (lines 225–234): “To assess the relationship between participants’ performance on the pre- and post-test OSVE and prior POCUS training, participants were divided into two groups: those with prior POCUS training and those without. No difference was observed when comparing pre-test scores (T = 0.643; p = 0.535), post-test scores (T = -0.0570; p = 0.956), and the change between pre and post-test scores (T = 0.522; p = 0.616) among both groups. To evaluate whether prior hands-on POCUS experience—measured by the number of times participants reported having performed POCUS before the study—was associated with the change in OSVE scores, we applied the Spearman rank-order correlation. No significant relationship was found between the number of prior POCUS procedures and score variation (R < 0.001; p = 1.000).”

These findings were also addressed in the Discussion (lines 476–485): “This study found no statistically significant differences in OSVE performance between participants with and without prior formal POCUS training, either in the pre-test, post-test, or in the variation between them. Likewise, no correlation was observed between the number of prior POCUS procedures performed and changes in test scores. These findings suggest that previous exposure to POCUS—whether formal or practical—was not associated with better performance in this structured assessment. A possible explanation is that the OSVE may evaluate broader skills such as clinical reasoning and integration of ultrasound findings into decision-making, which may not be fully developed through prior training alone. Additionally, variability in the type and depth of previous training, as well as the small sample size, may have limited the detection of differences.”

Learning Curve and Supervision

Comment: The authors' own finding that "No significant difference was found between pre-test and post-test performance (T = 105.00; p = 0.970), indicating that our single short-duration POCUS training did not improve participants' performance in image interpretation" reinforces the understanding that POCUS, despite its short learning period, typically requires supervision and multiple hands-on exposures for proficiency in image interpretation. We did not find any mention of how the POCUS findings were verified.

Response: We thank the reviewer for this important observation. We realize that our original wording may have caused ambiguity regarding what was meant by "participants' performance in image interpretation." The phrase refers specifically to participants’ performance in interpreting ultrasound images presented in the Objective Structured Video Exam (OSVE), not the interpretation of POCUS images they acquired during the study. To clarify this, we revised the sentence in the Results section (lines 463–465) as follows: “No significant difference was found between pre-test and post-test performance (T = 105.00; p = 0.970), indicating that our single short-duration POCUS training did not improve participants' performance in image interpretation on the OSVE.”

We appreciate the reviewer’s highlighting of these points, which helped improve clarity and transparency in our manuscript.

Pre- and Post-Training Test Participation

Comment: The limited participation rates for the pre- and post-training objective structured video examinations (OSVE) (63% and 53% respectively) could potentially dilute the observed impact of the training on participants' knowledge. This limitation should be acknowledged when discussing the effectiveness of the training.

Response: We thank the reviewer for this observation. We agree that the limited participation rates in the pre- and post-training OSVE assessments may have influenced the measured impact of the training intervention. To address this, we have explicitly acknowledged this limitation in the Discussion section (lines 471–473), adding the following statement: “Additionally, the limited participation rates in the pre- and post-training OSVE—63% and 53% respectively—may have diluted the observed impact of the training on participants' knowledge.”

This acknowledgment emphasizes the potential influence of incomplete data on the evaluation of the training’s effectiveness.

Verification of Diagnostic and Management Changes

Comment: The methodology for verifying the accuracy of diagnosis and management decision changes needs to be clearly explained. How were these changes objectively assessed and confirmed?

Response: We thank the reviewer for highlighting this important point. We agree that the manuscript did not clearly specify the verification process for the accuracy of diagnostic findings and management decisions influenced by POCUS. To address this, we have added a clarifying statement in the Methods section (lines 151–153), as follows: “Since POCUS images were not recorded, it was not possible to independently verify the accuracy of the diagnostic findings or the clinical management decisions informed by them.”

This addition acknowledges the study limitation regarding objective verification and improves transparency about the methodology used.

Timing of Feedback

Comment: The timing of feedback regarding the usefulness of POCUS in clinical practice is crucial. If the feedback was collected immediately post-training, it may not accurately reflect the sustained utility or impact. Please clarify when this feedback was gathered.

Response: We thank the reviewer for pointing out this important issue. Upon review, we recognized that the original manuscript did not clearly specify the timing of the participants’ impressions and satisfaction survey. To clarify, we removed the paragraph stating that the survey was completed immediately post-training. We replaced it with a more accurate description in lines 160–163: “Upon study completion, participants completed a survey (Fig 1) assessing their satisfaction with POCUS training, its usefulness and relevance in clinical practice, confidence in using the method, and any challenges encountered with the equipment (S3 appendix C).”

This revision reflects that feedback was collected at the end of the study, thus capturing a more comprehensive perspective on the sustained utility and impact of POCUS in clinical practice. Accordingly, we updated Figure 1 to represent this change. Additionally, due to the change in appendix citations, we reversed the order of appendices B and C to maintain consistency in the text.

We believe these clarifications enhance transparency and address the reviewer’s concern about the timing of feedback collection.

Self-Confidence Evaluation

Comment: It appears that self-confidence while using POCUS in clinical scenarios was evaluated in only 10 out of 19 respondents (50% of the study subjects). This smaller sample size for confidence evaluation should be highlighted as a limitation.

Response: We appreciate the reviewer’s comment regarding the sample size for the self-confidence evaluation. We agree that assessing self-confidence in only 10 out of 19 participants (53%) represents a limitation that restricts the generalizability and strength of any conclusions drawn from this analysis. Accordingly, we have explicitly acknowledged this limitation in the Discussion section (lines 496–499):

“Although no significant correlations were found between prior formal training and self-confidence, it is important to highlight that self-confidence in using POCUS in clinical scenarios was assessed in only 10 out of 19 respondents (53%), which limits the strength of conclusions regarding confidence and should be considered a limitation.”

This addition ensures transparency regarding the interpretation of our findings related to participants’ self-confidence using POCUS in clinical practice.

Results Presentation

Redundancy in Reporting

Comment: Avoid repeating numerical values in the text if they are already clearly presented in tables or figures (e.g., Figure 2, 3, 4, Table 2).

Response: We thank the reviewer for this important suggestion regarding the clarity and conciseness of our results presentation. In response, we have revised the manuscript to reduce redundancy between the text and the tables/figures, as follows:

• For Table 1, we removed detailed numeric descriptions from the text and instead provided a concise summary statement (lines 202–203):

“Participant demographics and characteristics are summarized in Table 1.”

• Similarly, for Table 2, we streamlined the description (lines 339–341):

“To assess participants’ performance over the course of the study, we compared the first and last POCUS clinical impact forms completed by each participant. Final forms were completed by 73.7% of them. Summary data are presented in Table 2.”

• For Figures 2 and 3, we opted to avoid repeating detailed data in the text and included a general reference to these figures (lines 254–255):

“Among the 104 forms, POCUS was used by participants in various modalities and for multiple clinical purposes, as summarized in Figs 2 and 3, respectively.”

• However, for Figures 1, 4, and 5, we chose to maintain both textual descriptions and the figures themselves. We believe this approach enhances the reader’s understanding of the study methods and results, as the textual explanations complement the visual data presentation.

These revisions improve the manuscript’s readability by eliminating unnecessary repetition while preserving clarity and comprehension.

POCUS Utilization Data

Comment: While the presentation of organ evaluations (lines 237–246) provides some detail, a more impactful representation of POCUS clinical uses could be achieved by focusing on the 167 items for clinical uses depicted in lines 251–258, as these directly address the clinical questions POCUS is designed to answer.

Response: We appreciate the reviewer’s insightful suggestion to emphasize the clinical uses of POCUS over organ-specific evaluations, as this better reflects the practical utility of the method in answering bedside clinical questions. In accordance with this recommendation, we have revised the discussion section by removing the paragraph originally between lines 501 and 506, which described the frequency of organ evaluations. This paragraph was replaced by a more concise statement (lines 451–453):

“POCUS was applied across diverse clinical contexts, including cardiopulmonary, urinary, and hemodynamic assessments, underscoring its versatility in hospital-based practice.”

This change allows us to focus the discussion on the broad clinical applicability of POCUS, highlighting its role in guiding patient care decisions rather than solely detailing the anatomical areas assessed. We believe this adjustment better captures the essence of POCUS utilization and aligns with the reviewer’s recommendation.

Time Efficiency Analysis

Comment: In the "Time spent per number of modalities" assessment, the results section only states "The total time spent using POCUS was 935 minutes, with a mean value of 9 minutes per patient and 5.3 minutes per modality performed." It would be valuable to evaluate the effect of prior POCUS exposure on time efficiency.

Response: We appreciate the reviewer’s suggestion to further explore the influence of prior POCUS exposure on time efficiency. Accordingly, we conducted an additional analysis, which has been incorporated into the Results section (lines 264–273). Specifically, we assessed the total time spent per modality in relation to participants’ previous POCUS experience, measured both by self-reported number of exams performed before the study and formal training status.

Our findings in

---

## [Decision Letter · Decision Letter 1]

25 Aug 2025

Dear Dr. Dias Souza,

Thank you for submitting your manuscript to PLOS ONE. After careful consideration, we feel that it has merit but does not fully meet PLOS ONE’s publication criteria as it currently stands. Therefore, we invite you to submit a revised version of the manuscript that addresses the points raised during the review process.

We look forward to receiving your revised manuscript.

Kind regards,

Biswabandhu Jana, Phd

Academic Editor

PLOS ONE

Journal Requirements:

Reviewers' comments:

Reviewer's Responses to Questions

**Comments to the Author**

Reviewer #3: (No Response)

Reviewer #4: All comments have been addressed

2. Is the manuscript technically sound, and do the data support the conclusions?

Reviewer #3: No

Reviewer #4: No

3. Has the statistical analysis been performed appropriately and rigorously?

Reviewer #3: Yes

Reviewer #4: No

4. Have the authors made all data underlying the findings in their manuscript fully available?

Reviewer #3: Yes

Reviewer #4: Yes

5. Is the manuscript presented in an intelligible fashion and written in standard English?

Reviewer #3: Yes

Reviewer #4: Yes

Reviewer #3: Congratulations to the authors of the manuscript titled 'Assessing the clinical relevance of point of care ultrasound for hospitalists: impact on clinical reasoning and decision making'. The authors have done an excellent job of responding to the comments from the previous reviewers. I must specify that I was not part of the previous set of review. The authors have done a good job of presenting the data and discussing it. The language is clear, and illustrative of the work that has been done. The work presentation itself is generally high quality and i want to congratulate the authors for this.

However, I do have significant concerns with the study design and the conclusions that are drawn from them. Please see my issues here:

1. POCUS education for 120 minutes without a follow up period of verification of competency or progression of skill.

2. After completion of the education, the participant images were not reviewed for accuracy, completeness or appropriate interpretation in the clinical context.

Due to these two issues, the current conclusion is that 120 mins of education without verification of skills is appropriate for patient management and skill application into patient care. This is extremely concerning for me because there is a likelihood of harm reaching to the patient. I am not comfortable with that conclusion and possible outcome.

Due to these limitations, I personally cannot suggest that the manuscript is accepted for publication.

Reviewer #4: (No Response)

**Do you want your identity to be public for this peer review?** For information about this choice, including consent withdrawal, please see our Privacy Policy

Reviewer #3: No

Reviewer #4: No

---

## [Author Response · Author response to Decision Letter 2]

6 Oct 2025

Response to Reviewer #3

We thank the reviewer for the insightful comments and for highlighting essential patient-safety considerations related to POCUS training and verification. We revised the manuscript to clarify the scope of our educational intervention and the interpretability of our findings.

Comment 1:

“POCUS education for 120 minutes without a follow up period of verification of competency or progression of skill.”

We agree that a brief session does not establish technical competency. In the revised Methods, we clarify that the 120-minute session functioned as a brief orientation to align participants on basic Point-of-care Ultrassound (POCUS) concepts, terminology, and device operation, and was not designed to establish or verify technical competency. No follow-up skill verification or assessment of progression was performed, and the session was not analyzed as an intervention. Additionally, to streamline the manuscript and keep focus on the primary objectives, we removed the Objective Structured Video Examination (OSVE) and all related pre/post analyses, as well as the exploratory comparison between the first and last POCUS forms and the derived efficiency metric (“time spent per number of modalities”). These changes prevent secondary analyses of educational (OSVE) and process measures from overshadowing our primary per-encounter descriptions of how clinician-directed POCUS related to diagnostic hypotheses, management decisions, and confidence during routine care.

Comment 2:

“After completion of the education, the participant images were not reviewed for accuracy, completeness or appropriate interpretation in the clinical context.”

We agree and have made this limitation explicit in both the Methods and the Limitations sections, and we also reiterate it in the Conclusion to ensure a cautious interpretation of the findings.

Comment 3:

“Due to these two issues, the current conclusion is that 120 mins of education without verification of skills is appropriate for patient management and skill application into patient care. This is extremely concerning for me because there is a likelihood of harm reaching to the patient. I am not comfortable with that conclusion and possible outcome.”

We fully agree and have revised the entire manuscript to adopt a more cautious framing. The revised text emphasizes feasibility and perceived influence on clinical reasoning, rather than suggesting effectiveness of training. We now state that stronger study designs with validated protocols and competency assessment are required to confirm these findings.

Response to Reviewer #4

Comment 1:

“The chosen study design, however, requires clarification and more precise description. The manuscript currently refers to the study in broad terms, but given the structure — participants undergoing baseline assessment, receiving an educational intervention (POCUS training), and then being reassessed with both OSVE and questionnaires — this aligns most accurately with a prospective quasi-experimental before-and-after design. This classification is important because the study does involve an intervention and measures outcomes before and after its implementation, but it does not include randomization or a parallel control group. Calling it a ‘descriptive prospective study’ would be misleading, as descriptive studies do not assess causal effects or intervention impact. Properly labeling the design as quasi-experimental strengthens methodological transparency and allows readers to appropriately interpret the results.”

We thank the reviewer for the careful assessment. In response, we removed all pre/post analyses related to the educational session, including any Objective Structured Video Examinations (OSVE) based comparisons, to keep the study focused on our primary objective. The educational session is now presented exclusively as introductory orientation (not an analyzed intervention). Importantly, the OSVE is described only as exposure to standardized clinical vignettes containing Point-of-care Ultrassound (POCUS) images to align participants’ familiarity with image-based content; no OSVE scores or pre/post contrasts enter our analyses. With these changes, the manuscript no longer evaluates intervention effects and does not estimate causal impact. The study therefore fits a prospective descriptive observational design. We also added clarifying language to prevent any implication of a quasi-experimental before-and-after framework.

Comment 2:

“I have concerns regarding the measurement of two key process variables: (a) time spent performing the POCUS exam, and (b) quality of images obtained. According to the data collection form, both variables were captured through participant self-report rather than objective measurement. For time spent, it is unclear whether this was based on a stopwatch or structured observation; as presented, the data seem to rely entirely on participants’ subjective estimation. Similarly, the assessment of image quality was based on participants’ own descriptions without saving or independently reviewing the images. This introduces a high risk of recall bias and self-report bias. Moreover, not storing images for subsequent blinded quality review represents a missed opportunity to validate this outcome objectively. These limitations should be explicitly acknowledged in the manuscript, as they constrain the strength of the conclusions regarding feasibility and quality of POCUS implementation.”

We thank the reviewer for these important observations. In the revised manuscript, we clarified in the Methods that both process variables were self-reported rather than objectively measured. Specifically, we now state that exam duration was reported as an estimated time by the participant, not measured with a stopwatch or structured observation. Similarly, we emphasize that image quality was based solely on participants’ self-reported perceptions. In addition, we made it explicit that POCUS images were not recorded, which precluded independent verification of acquisition quality, diagnostic accuracy, or related management decisions.

We also revised the Discussion to explicitly acknowledge these issues as limitations. We highlight that reliance on participants’ estimations introduces a risk of recall bias, while self-reported image quality without external review introduces self-report bias. Furthermore, we note that not storing images prevented objective validation of findings, interobserver reliability assessment, or concordance with gold-standard criteria. These aspects are now clearly recognized as factors that constrain the strength of our conclusions regarding feasibility and quality of POCUS implementation.

Comment 3:

“In addition, the manuscript does not provide sufficient information on the development and validation of the data collection forms. It is unclear whether the instruments underwent a systematic process of construction (e.g., based on literature review, expert consultation) and whether any procedures to establish evidence of content validity were performed. Without such information, it is difficult to assess whether the forms adequately captured the intended constructs (e.g., diagnostic hypotheses, management changes, confidence levels, time spent, and image quality). The absence of reported validity evidence represents an important limitation, as the accuracy and interpretability of the findings rely heavily on the adequacy of these instruments.”

We thank the reviewer for this valuable observation. In the revised manuscript, we clarified in the Methods that the data collection form was collaboratively developed by the authors, informed by prior studies in internal medicine/hospital medicine POCUS, and tailored to the specific objectives of this study. We also make explicit that the instrument was not formally validated and that no psychometric procedures (e.g., content validity assessment, inter-rater reliability testing) were performed.

Furthermore, we revised the Discussion to explicitly acknowledge this limitation. We highlight that, because the form was author-developed and not validated, the accuracy and interpretability of the constructs captured (diagnostic hypotheses, management changes, confidence levels, exam duration, and image quality) may be constrained. This limitation is now clearly stated alongside other methodological caveats.

Comment 4:

“In addition, it is unclear whether the POCUS training and subsequent examinations followed a standardized protocol, such as the RUSH protocol. The data collection form included the following examination categories: cardiac, pulmonary, abdominal, inferior vena cava, femoral and popliteal veins, bladder, and ‘other’. The RUSH protocol encompasses cardiac function and contractility assessment, evaluation of intravascular volume status via the inferior vena cava, identification of free fluid in the abdomen, lung assessment for pneumothorax or pulmonary embolism, and lower limb DVT screening. Given this overlap, it should be explicitly stated whether the investigators adopted the RUSH protocol in its entirety, used a modified version, or employed another structured approach. If the RUSH protocol was not used, the authors should clearly describe which anatomical regions and clinical targets were prioritized in both training and clinical application."

We thank the reviewer for raising this point. In the revised Methods, we clarified that neither the educational session nor the subsequent clinical examinations followed a standardized protocol such as RUSH. Examinations were performed entirely at the discretion of each physician, guided by the specific clinical question in each encounter.

The educational session provided a broad, introductory overview of POCUS for hospitalists, covering lungs/pleura, heart, abdomen, inferior vena cava, femoral/popliteal veins, and bladder, as well as basic equipment handling and image optimization. This was intended as an orientation, not as training in a fixed algorithm. The overlap between our data collection categories and components of the RUSH protocol reflects the fact that these organ systems are commonly relevant in hospital medicine POCUS, rather than adoption of RUSH or any other structured pathway.

Comment 5:

“The manuscript states that responses to open-ended questions were kept in their original format, but it does not describe the procedures adopted for their analysis. It remains unclear whether these responses were analyzed qualitatively (e.g., through thematic or content analysis), coded systematically by independent reviewers, or simply presented as anecdotal remarks. Without a clear description of the analytic approach, the contribution of these data to the study’s findings is difficult to assess. The authors should specify the method used to analyze the open-ended responses, including whether any coding framework, inter-rater reliability, or software tools were applied. If the responses were not systematically analyzed, this should be acknowledged as a limitation.”

We thank the reviewer for this important observation. In the revised manuscript, we clarified in the Methods that open-ended responses and relevant closed-ended items were transcribed into a structured Excel sheet and coded by the principal investigator for each form. Binary variables were created to capture changes in diagnostic hypotheses, diagnostic confidence, addition or exclusion of alternative diagnoses, incidental findings, and management changes. For diagnostic confidence, responses were recoded into three categories (Likert 1–2, 3, and 4–5) to define shifts across confidence levels. These derived variables were analyzed descriptively as categorical data. Open-text responses were also reviewed to extract brief contextual information; however, no formal qualitative analysis was performed, and no independent coding, inter-rater reliability procedures, or software-assisted thematic analysis were applied.

We also revised the Discussion to explicitly acknowledge this as a limitation. Specifically, we note that open-ended responses were categorized by a single investigator without independent review, coding framework validation, or reliability assessment, which may have introduced subjective bias. This caveat is now clearly presented alongside the other methodological limitations to aid readers in interpreting the findings.

Comment 6:

“The sample size rationale estimates a single post-POCUS proportion with 95% confidence and 10% margin of error, assuming p=0.20, yielding ≈62 forms. This is acceptable for estimating a single proportion. However, because multiple forms were submitted by the same hospitalists, observations are clustered. The calculation should therefore include a design-effect inflation based on the average forms per clinician and an assumed ICC. Using 5.5 forms/physician and plausible ICCs (0.05–0.10) increases the requirement to approximately 76–90 forms. The 104 forms collected likely satisfy this, but the manuscript should explicitly report the clustering adjustment (DEFF and ICC). In addition, the finite-population correction used should reflect the finite number of eligible encounters in the unit rather than the number of internists in the state; otherwise, FPC should be omitted.”

We thank the reviewer for this important observation. We agree that multiple forms submitted by the same physician introduce clustering, requiring adjustment. In the revised manuscript, we now explicitly report this. Because it is not possible to determine the finite number of eligible encounters within the Transitional Care Unit, we omitted the finite-population correction, as recommended by the reviewer, and applied the single-proportion formula assuming an infinite population.

In addition, we incorporated the reviewer’s recommendation to include a design-effect adjustment based on the average number of forms per physician and an assumed intraclass correlation coefficient (ICC). Since there are no prior studies to inform a precise ICC value in this context, we opted for a conservative estimate of ICC = 0.10. Using the observed average of 5.5 forms per hospitalist in our dataset, this adjustment increased the required sample size from 62 to approximately 90 forms. Our final dataset of 104 forms exceeded this threshold, thus satisfying the adjusted requirement.

Comment 7:

“The study demonstrates positive trends but does so with methodological fragility. The results should be framed more cautiously, emphasizing feasibility and potential utility rather than efficacy. Stronger study designs (e.g., randomized controlled trials or controlled before-and-after studies with validated outcome measures and objective performance data) would be necessary to substantiate these findings”

We fully agree and have revised the entire manuscript to adopt a more cautious framing. The revised text emphasizes feasibility and perceived influence on clinical reasoning. We now state that stronger study designs with validated protocols and competency assessment are required to confirm these findings.

Final response to reviewers

In addition to the revisions made in direct response to the reviewers’ comments, we implemented a few further improvements to enhance clarity and readability.

First, we edited the following sentence to improve the fluency of the text:

“Upon study completion, participants completed a survey (Fig 1) assessing their satisfaction with POCUS training, its usefulness and relevance in clinical practice, confidence in using the method, and any challenges encountered with the equipment (S3 Appendix C)” to: “Upon study completion, participants filled out a survey (Fig 1) assessing their satisfaction with POCUS training, its usefulness and relevance in clinical practice, confidence in performing POCUS, and any equipment-related challenges (S3 Appendix C).”

Second, because exam-time data did not follow a normal distribution according to the Shapiro–Wilk test (as described in the Methods), we replaced the mean with the median and interquartile range for the per-patient time: “The total time spent using POCUS was 935 minutes, with a median of 10 minutes per patient [5.00; 10.00].”

Third, to facilitate interpretation, we added Table 2 — POCUS Influence on Diagnostic Reasoning and De

---

## [Decision Letter · Decision Letter 2]

19 Nov 2025

Assessing the clinical relevance of point-of-care ultrasound for hospitalists: influence on clinical reasoning and decision-making

PONE-D-25-23495R2

Dear Dr. Dias Souza,

We’re pleased to inform you that your manuscript has been judged scientifically suitable for publication and will be formally accepted for publication once it meets all outstanding technical requirements.

Kind regards,

Biswabandhu Jana, Phd

Academic Editor

PLOS ONE

Additional Editor Comments (optional):

Reviewers' comments:

Reviewer's Responses to Questions

**Comments to the Author**

Reviewer #5: All comments have been addressed

2. Is the manuscript technically sound, and do the data support the conclusions?

Reviewer #5: Yes

3. Has the statistical analysis been performed appropriately and rigorously?

Reviewer #5: Yes

4. Have the authors made all data underlying the findings in their manuscript fully available?

Reviewer #5: Yes

5. Is the manuscript presented in an intelligible fashion and written in standard English?

Reviewer #5: Yes

Reviewer #5: The manuscript presents a well-structured and technically sound study. The objectives are clearly defined, and the methodology is appropriate for the problem being addressed. The authors provide a solid theoretical background, and the proposed approach is supported with adequate analysis and relevant comparisons. The results are presented in a clear and organized manner, demonstrating the effectiveness of the proposed method. Overall, the contribution is meaningful, and the technical content meets the standards expected for publication.

**Do you want your identity to be public for this peer review?** For information about this choice, including consent withdrawal, please see our Privacy Policy

Reviewer #5: No

---

## [Editor Report · Acceptance letter]

PONE-D-25-23495R2

PLOS One

Dear Dr. Dias Souza,

I'm pleased to inform you that your manuscript has been deemed suitable for publication in PLOS One. Congratulations! Your manuscript is now being handed over to our production team.

Kind regards,

on behalf of

Dr. Biswabandhu Jana

Academic Editor

PLOS One